# Fire Risk Assessment of Combustible Exterior Cladding Using a Collective Numerical Database

**Timothy Bo Yuan Chen [1], Anthony Chun Yin Yuen [1,*] , Guan Heng Yeoh [1,2], Wei Yang [1,3] and Qing Nian Chan [1]**

[1]   School of Mechanical and Manufacturing Engineering, University of New South Wales, Sydney, NSW 2052, Australia; timothy.chen@unsw.edu.au (T.B.Y.C.); g.yeoh@unsw.edu.au (G.H.Y.); weyang@ustc.edu.cn (W.Y.); qing.chan@unsw.edu.au (Q.N.C.)

[2]   Australian Nuclear Science and Technology Organisation (ANSTO), Locked Bag 2001, Kirrawee DC, NSW 2232, Australia

[3]   Department of Chemical and Materials Engineering, Hefei University, Hefei 230601, China

*   Correspondence: c.y.yuen@unsw.edu.au; Tel.: +61-2-9385-5697

**Abstract:** Recent high-profile building fires involving highly-combustible external cladding panels in Australia as well as Dubai, China, and the United Kingdom have created a heightened awareness by the public, government, and commercial entities to act on the risks associated with non-compliant building structures. In this paper, a database of fire events involving combustible aluminium composite panels was developed based on (i) review of relevant major fire events in Australia and other countries, and (ii) numerical simulation of the ignitability, fire spread, and toxic emissions associated with composite panels. Through the application of large-eddy-simulation (LES)-based computational fire field models, the associated risks for a standardized two-storey building with external cladding was considered in this study. A total of sixteen simulation cases with different initial sizes of the fire and different air cavity widths in the exterior cladding assembly were examined to investigate the tolerable situations and their influences. It was discovered that for most cases, with an initial fire size greater than $400 \ \mathrm{kW/m^{-2}}$, the fire will spread from the first to second floor before the allowed egress time period.

**Keywords:** fire risk assessment; combustible building materials; high-rise buildings; large eddy simulation; pyrolysis

## 1. Introduction

The rapidly increasing utilisation of advanced lightweight materials, including light alloys, polymers, and fibre-reinforced composites that are highly flammable poses significant fire risks impacting people, environment, and the economy. They can be often found in exterior cladding systems, otherwise known as exterior insulation finishing systems (EIFS) or external thermal insulation composite systems (ETICS). These systems are designed to be cost effective solutions for thermal insulation, weather resistance, and aesthetic external wall finishes. In Australia, the most basic exterior cladding system consists of:

1.   An insulation layer, often a polymer such as polystyrene (EPS), polyisocyanurate (PIR) or polyurethane (PU);
2.   A surface finish layer that can be a surface coating or a sandwich panel (such as an aluminium composite panel (ACP)).

Composite panels or sandwich panels are made of a thin outer metal skin of steel or aluminium and cores of insulating material which often include highly-combustible expanded polystyrene (EPS),

polyurethane (PUR), and sometimes polyethylene (LDPE) and mineral fibre. In recent years, owing to alarming concerns of significant fire incidents caused by the burning of sandwich panels such as the Grenfell Tower Fire and the Dubai Tower Fires, it was recently discovered that polymer materials within these panels including EPS, PUR, and LDPE were the root causes of these fires. These events have created a heightened awareness by the public and have propelled governmental authorities and commercial entities to act on the risks associated with the non-compliance of such structures that have been erected in the building and construction landscape. Therefore, it is essential to understand the fire behaviours of exterior cladding systems and how different materials and their configuration effect the flammability of that system. Currently, there are many different types of exterior cladding systems and the complexity varies according to the number of layers. These layers often consist of polymeric materials, therefore the potential for a flammable cladding system increases with the increase in complexity of the system [1]. Other than the high-flammable core found in aluminium composite panels, there are many other factors that influence the fire safety of exterior cladding systems. These include the width of the cavity between the insulation and the external panels, the types of insulation material, the installation of fire barriers in between levels, and the structural weaknesses of joints and connection between individual panels that deteriorate with high temperature. Unfortunately, the relevant legislation and building codes have yet to catch up with the requirements for assessing the fire risks involved in these buildings and many key aspects of exterior facade flammability are not well understood [1,2]. Therefore, it is of great importance to develop a systematic approach to evaluate the risks for existing and ongoing development of combustible cladding materials (i.e., ACPs) that could be applicable to a wide range of building configurations.

In this study, a review of past major fire incidents from 1990 to the present is provided. Furthermore, a holistic methodology to analyse the fire risk of non-compliant buildings is proposed according to building code (AS 4391-1999). Finally, this method will be examined by means of a full-scale numerical fire simulation to investigate its viability.

## 2. Review of Past Fire Cases

The history of fire cases involving combustible external composite panels stretches back many decades. Composite panels were first developed as a cost-effective, lightweight building material that could be rapidly installed for external cladding or facades of industrial buildings. Through considerable development over the past few decades, these panels are now widely used across a vast variety of buildings. The main advantage of composite panels is that they are inexpensive, can be easily cut and shaped in any size or dimension, are lightweight, and have excellent insulation characteristics. The products also come with a wide variety of surface finishes to suit architectural designs. The issue of combustible composite panels now concerns both private residences and commercial offices and factories.

### 2.1. Knowsley Heights Fire, Liverpool UK, 1991

One of the first historical cases of an external cladding fire occurred in Knowsley Heights, a residential building which was refurbished with additional thermal insulation to the external walls. The fire started in a rubbish compound outside the building and ignited the external cladding system, which spread rapidly across the face of the building [3]. This incident resulted in the introduction of horizontal cavity barriers at each floor to prevent other similar incidents from occurring again.

### 2.2. Garnock Court Fire, Scotland, 1999

The Garnock Court fire occurred on 11 June 1999, a 14-storey block of flats in Irvine, Scotland, which resulted in one death and four injuries. The Garnock Court fire in Scotland is a significant incident in history because it led to a significant change in building regulations for Scotland in 2005 which included "resistance to the spread of fire" as a functional standard for buildings [4].

### 2.3. Television Cultural Centre Fire, China, 2009

The Television Cultural Centre (TVCC) fire occurred on 9 February 2009, the night before Chinese New Year (Figure 1). The incident caused the death of one firefighter and seven injured civilians. The tower's northern and southern facades were installed with glass curtain walls, while the east and west claddings featured metal panels and strips made with titanium–zinc alloy. The fire was initiated on the roof by fireworks. The sparks from the fireworks on the roof penetrated the metal panels and ignited the insulation materials (extruded polystyrene (XPS) foam) and waterproof sheets (EPDM rubber). There was a cavity between the metal panel and the insulation layers. It was reported that the melting and burning droplets of XPS flew down the facades. Combined with strong winds, the tower was entirely in flames within less than 20 min [5]. The extensive use of highly-combustible insulation and the large cavities without breaks are believed to have contributed significantly to the downward spread of the fire from the upper to lower floors.

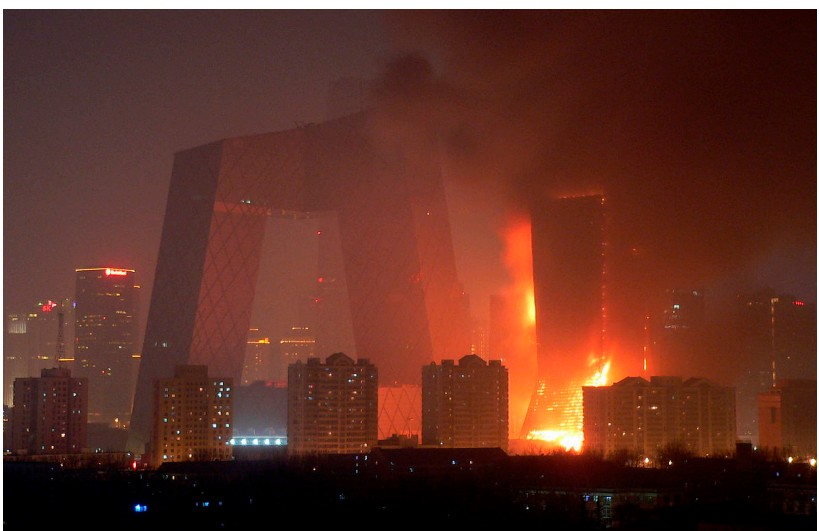

**Figure 1.** The north wing of the new Television Cultural Centre (TVCC) tower on fire, 09 February 2009. Photograph by Wing, distributed under CC-BY license.

### 2.4. Shanghai Apartment Fire, China, 2010

In 2010, a fire occurred in a 28-storey residential building in Shanghai, causing 58 casualties and injuring 71. It was one of the most catastrophic fire incidents in history involving combustible cladding material. At the time, the building was undergoing renovations for the installation of exterior wall insulations (polyurethane (PU) foam); the scaffolding was decked with wood and bamboo, a typical scaffolding material in China [6]. It is believed that the fire was caused by welding sparks which ignited the insulation chips or the wood and bamboo scaffolding on the 9th/10th floor [7]. The insulation materials were highly flammable with good access to air supply, and the fire spread externally at a rapid rate. It was reported that the fire spread to the roof in 4 minutes, and within 14 min the entire north-facing facade was utterly burned out [8]. The fire also spread into several apartment rooms. While the building was equipped with automatic sprinklers systems from the 1st to the 4th floors, they were completely ineffective in controlling the spread of the fire on the external walls.

### 2.5. Tecom Building Fire, Dubai, 2012

The Tecom building fire in Dubai injured two people and damaged nine floors of the building. The building was installed with aluminium composite panels with a polyethylene core. High amounts of burning debris consisting of twisted metal rods and sheets fell onto the streets. It damaged five vehicles and left bystanders on the road with minor burn injuries [9].

*2.6. Lacrosse Building Fire, Australia, 2014*

The Lacrosse Building Fire occurred on the 24 November 2014 in a 23-storey apartment building in Melbourne, Australia. The fire started via an unextinguished cigarette disposed in a plastic container on the 6th floor balcony which spread to the timber table top. Aluminium composite panels with a polyethylene core were installed on the side wall of the balcony. Once the fire ignited the external wall cladding, it rapidly spread vertically up the building. Due to the rapid fire spread and penetration into the internal parts of the building over many levels, the entire building (with over 400 occupants) were evacuated. In addition, the fire also caused the fire alarm warning system to fail on some of the levels and firefighters were forced to enter every level and alert the occupants to ensure total evacuation. Fortunately, no casualties or serious injuries occurred during this incident.

*2.7. The Torch, Dubai, 2015*

The Torch is a 79-storey skyscraper, one of the world's tallest residential towers. The fire started on the 50th floor by what is believed to be a cigarette or Shisha coal left on the balcony. It was one of a series of skyscraper fires that occurred in a relatively short period time between 2015–2016 [10]. More than 40 floors were burning on one side of the building and large quantities of flaming materials fell from the high-level fire which started a secondary fire at lower levels [11]. The burning debris was also carried by strong winds and littered surrounding streets. This was particularly hazardous for the densely populated city of Dubai.

*2.8. The Address, Dubai, 2016*

The Address was another skyscraper fire that occurred in Dubai which caused 15 injuries. The 63-storey building caught fire via a short circuit in one of the external floodlights installed on a ledge formed by horizontal cladding panels between the 14th and 15th floor [12]. The building was installed with aluminium composite panels with a polyethylene core, which was blamed for the rapid fire spread across the building.

*2.9. Grenfell Tower, UK, 2017*

The Grenfell Tower fire broke out on 14 June 2017 in the 24-storey Grenfell Tower block of public housing flats in North Kensington, West London, United Kingdom (Figure 2). The London Metropolitan Police confirmed that 80 people died as a result. The event will go down in history as one of the most horrific fire disasters alongside the Shanghai apartment fire in 2010. Prior to the incident, Grenfell Tower underwent a major renovation on the exterior of the building which included new windows, a heating system, and the installation of a new exterior cladding for insulation and rainscreen [13]. The fire is believed to have been started from a refrigerator in a 4th floor apartment kitchen. The residents were in the apartment at the time and called the fire brigade. Despite firefighters arriving 6 min after the alarm, the fire managed to spread to the exterior cladding before the firefighters suppressed the kitchen fire. The flames spread at an alarming rate up the exterior cladding and the fire quickly became out of control. In addition, the exterior fire re-entered the building, trapping a significant percentage of residents inside the building [14].

The fires in the Grenfell Tower, and other high-rise buildings in Australia and internationally have all involved aluminium composite panels (ACPs), made of highly-combustible polyethylene (PE) core material. These incidents have exposed many of the flaws and negligence in current building regulations and fire safety protocols. In summary, the major risks of non-compliant buildings can be sub-divided into the following three categories:

- **Rapid Surface Propagation**—The fire characteristics of the material used in the cladding system significantly influence the rate of fire spread. All of the reviewed incidents show rapid fire spread on the external surfaces, generally developing into an uncontrollable blaze within 10–15 minutes. This highlight the significant fire risks of combustible cores for ACPs.

- **Cavities**—Cavities are included in the design of the cladding system for insulation purposes. It can also be formed by delamination or any structural movement caused by the fire. When the fire enters into the cavities, the fire can stretch up to 5–10 times the flame length to find oxygen for combustion. This phenomenon occurs regardless of the materials used and it enables the fire to spread rapidly and unseen within the cladding system. It is especially dangerous to firefighters as it creates hidden fires and toxic smoke build-up within the panels and sudden flashovers.
- **Fire re-entry**—Windows and other openings allow the fire to spread back into the building. As highlighted in many of the reviewed incidents, it can create secondary fires multiple floors away from the original fire location. Without timely intervention, this allows the flames to break out again, potentially trapping occupants that are located in between the two levels.

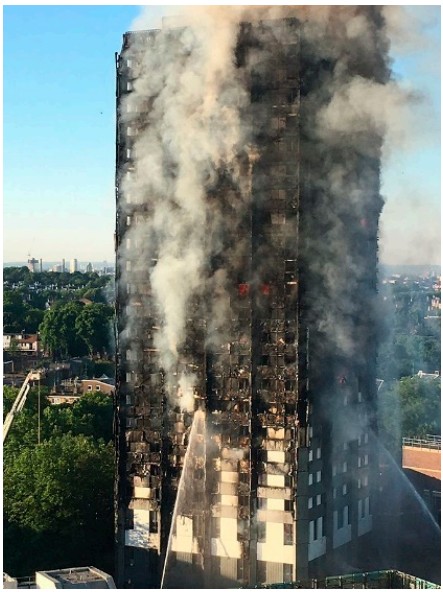

**Figure 2.** Image of the Grenfell Tower Fire. Photograph by Natalie Oxford, distributed under CC-BY license.

## 3. Provisional Regulations on Exterior Wall Claddings

In the Australian context, the Lacrosse Building fire and the Grenfell Tower Fire have highlighted the need for regulatory reforms regarding flammable cladding material. In addition, it also raises concerns regarding other building materials and the risk perception of emergency response personnel who confront these fire risks and make decisions whether to withdraw to protect their own safety. In June 2017, a Senate committee was established to investigate the use of cladding material on Australian buildings in the aftermath of the deadly Grenfell Tower fire [15]. The committee has identified a range of key issues that have contributed to the issue of non-compliance and non-conformity in building products in Australia. These include:

- **Increase in products imported from overseas**—There is a shift from local products to imported materials from international manufacturers, creating issues in third-party certification schemes and their reliability.
- **Reliability of certification documentation**—Increase in the number of cases of fraudulent or misleading material compliance documents which leads to non-compliant materials used in building constructions in Australia.
- **Inappropriate product substitution**—Aggressive cost-cutting in construction operations often leads to the substitution of inferior products that underperform compared to original specifications.

- **Clarity of material certification for conformity**—There is a lack of clarity in building codes in relation to flammable cladding material, which leads to a decrease in confidence in certificates of conformity issued under the Australian Building Codes (BCA).

Following the government inquiry, an inter-agency Fire Safety and External Wall Cladding Taskforce was established in all states to ensure that fire safety requirements involving external wall cladding are prioritised and adequately addressed. The taskforce proposed a process to manage the problem in four phases outlined in Figure 3. Essentially, the process presents an identification, assessment, and remediation workflow to deal with non-compliant aluminium cladding. The procedure starts with: (i) Data audit to identify buildings with combustible aluminium composite panels installed on the exterior cladding. (ii) Information and notice are then given to the owners and managers of the buildings identified in the audit. This can also include recommendations to check approvals and conduct fire safety assessments immediately. (iii) The taskforce will also undertake routine checks and require the owners to report back on the cladding on the residential buildings.

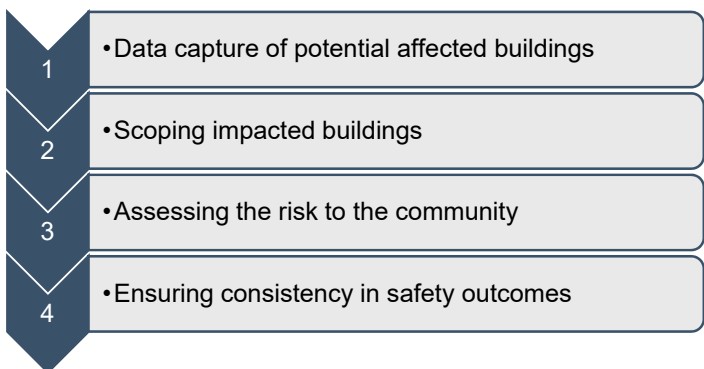

**Figure 3.** Four phase process for strengthening fire safety involving non-compliant building materials [16].

The cladding task force for each state recently released statistics on their audit for non-compliant building materials. In Victoria, an audit of 170 buildings found that 87 failed to comply with the BCA. In New South Wales, 1184 buildings were identified to have aluminium cladding. The Taskforce proposes to visit each of those buildings and has already identified 220 buildings, including 58 high-rise residential buildings necessitating further investigation [17]. Similar data have been reported in all the other states. Unfortunately, the relevant legislation and building codes have yet to catch up with the requirements for assessing the fire risks involved in these buildings. There is a lack of clarity in the National Construction Code for flammable cladding material. There are currently no existing fire safety regulations in Australia to promote the safe use of lightweight materials for external facades. Authorities have further proposed that the Taskforce should undertake product testing on commonly available external cladding products with the subsequent development of a materials library which would allow a rapid assessment and product classification for samples taken from government and private buildings. Furthermore, there were no actions taken to revise current firefighting protocols. With the recent development of bio-inspired flame-retardant coating solutions [18,19], this may possibly become an immediate measure to protect the existing claddings from fire hazards for a reasonable duration of 5–10 years.

With the urgent need to resolve the present fire risks of existing building products and develop economically viable solutions, it is paramount that an assessment tool be developed to evaluate underlying risks for existing and ongoing development of non-compliant materials on buildings. A deeper understanding of the associated risks is not only beneficial to the building occupants, but also to emergency responders, who risk their lives during the operation of fire events such as the Grenfell Tower. In the next sections, a generic assessment tool to evaluate fire risks for non-compliant materials on buildings is formulated. A case study is shown to demonstrate the concepts of the model and how it determines the risk levels for fire assessment.

## 4. Methodology

Currently, large-scale fire testing still remains the only possible route to gain knowledge about the flammability of exterior facades. However, these assessments are very costly, destructive, and often impossible due to many practical constraints. Even when a large-scale test is performed, it is currently done on a perfectly constructed system. In reality, the systems installed onto buildings may be vastly different to the testing standards [1]. Therefore, numerical simulations based on computational fluid dynamic (CFD) techniques are a cost-effective tool to bridge to the knowledge gap and explore the system sensitivity to some of the parameters such as the gap widths and material thickness.

The proposed fire risk assessment tool involves firstly establishing a fire database consisting of all the elements associated with the major causes and outcomes of exterior cladding fire from a combination of both past case studies and numerical simulations. According to the review of past fire cases, a typical configuration for an external cladding facade is constructed in a computational fire model. According to the numerical data predicted from the modelling framework, the tenable conditions for different scenarios can be identified systematically based on the existing building regulations provided in Australian and Fire Safety Code of Practice. Details of the risk criterions are outlined in Section 4.2. A collection of simulations was carried out to investigate the influences of the flaming condition, air cavity, and window opening entrainment towards the flame spreading across a two-storey building with external aluminium composite panel filled with polymer cores.

In this study, the numerical simulations were performed using the Fire Dynamics Simulator (FDS) version 6.0.2, which is a well-recognised software in the fire engineering community. There have been many successful case studies of fire scene reconstruction and forensics using FDS [20,21]. Furthermore, there are ongoing developments to extend the application to wildland fires by incorporating large-scale fire propagation models [22]. By considering the governing equations of the conservation laws (i.e., mass, momentum, energy and other variable properties), turbulence, combustion, radiation modelling, FDS is capable of simulating all essential behaviours and phenomena of non-premixed flame. It applies the large eddy simulation (LES) methodology and can simulate the temporal heat and mass transport, as well as gas species and smoke movement of fire plumes for low–speed flows [23]. In LES, the instantaneous fluctuation behaviours due to the turbulent mixing can be considered and fully-coupled with the combustion, soot, and radiation models [24,25]. The combustion model used in the current simulations is a mixing-controlled fast chemistry model based on the "mixed is burnt" assumption, in which the eddy dissipation concept [26,27] is applied for evaluation of the time-averaged chemical source term. Smoke generation is modelled base on a soot yield, which is defined as the mass of soot produced per mass of fuel. Although there are short-comings for minor chemical species predictions when considering reduced chemical kinetics, it is still effective for CO predictions with significantly improved computing efficiency [28]. Thus, for large-scale compartment fire, it is common to apply such approach to acquire an effective solution. For this study, a soot yield of 0.01 was adopted. This is a fair assumption for large scale fires in a large compartment where a more precise model is not a necessity. In the work conducted by Vigne and Węgrzyński [29,30], a wide range of soot yields ranging from 0.001 g/g to 0.178 g/g were tested to replicated various types of flames. It was reported that a range from 0.01 g/g to 0.015 g/g was suitable for polymeric composite materials such as polyethylene (PE). For turbulence modelling of small-scale eddies (i.e., length scale filter equivalent of the size of the grid), the Smagorinsky subgrid-scale (SGS) model [31] was adopted with a Smagorinsky constant of 0.2 while turbulent Schmidt and Prandtl numbers were prescribed as 0.5 respectively [32].

### 4.1. Model Configuration

According to the literature review, the typical configuration for an external cladding facade is illustrated in Figure 4. The computational domain consistsed of a two-level external cladding system. Each level was 2.4 m high and 2.4 m wide. Each level also included a 1.5 m × 1.0 m window located 0.9 m from the floor. A 0.4 m by 0.4 m fire was created at the first level interior. The external cladding

consisted of 50 mm of insulation (XPS, extruded polystyrene) and a 5 mm aluminium composite panel (4 mm of polyethylene core sandwiched between 0.5 mm aluminium). Air cavities of various distances were incorporated within the external cladding. The material properties were extracted from cone calorimetry and fire tests conducted at the Fire Rescue NSW Londonderry site [33]. The model was designed to emulate an interior fire spreading to the external cladding and re-entering to another floor through the windows. The measurements were taken in locations 1A, 1B, and 1C and 2A, 2B, and 2C for level 1 and level 2, respectively. These measurement locations were structured in a line across the window width, 0.5 m above the bottom edge. The exact locations of the measurement points are shown in Figure 4b.

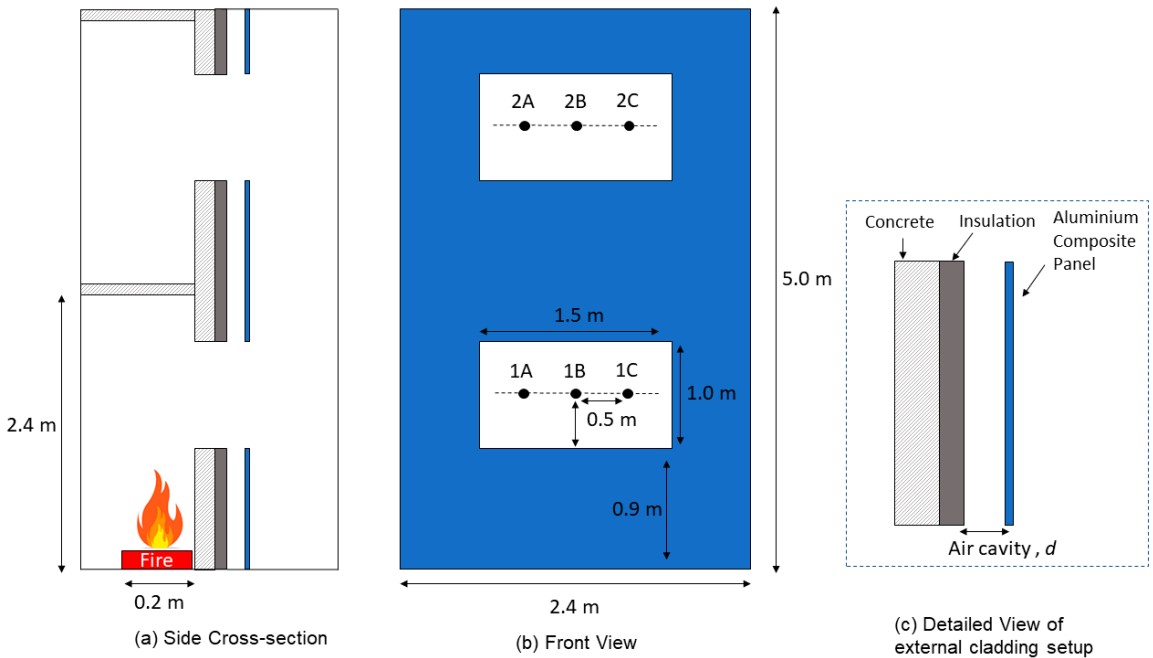

**Figure 4.** Model configuration.

As illustrated in Figure 5, the computational domain consists of a rectangular box with dimensions 2.4 m (W) × 4.5 m (L) × 5.0 m (H). The transparent faces represent opening boundary conditions and the yellow faces represent wall boundary conditions. A uniform mesh size of 0.02 m was applied with a total mesh number of 2,670,000. The uniformly size mesh system will provide a more stable solution since the aspect ratio is unity meaning that the order of error for partial derivative approximations were minimised. According to the characteristic length scale ($D^*$) analysis proposed by DiNenno et al. [34], the characteristic length of the mesh system adopted in this study were in the satisfactory range of $1/20 < R^* < 1/15$, where $R^* = \Delta l^* / D^*$ and $\Delta l^*$ is the overall mesh size applied in the computational domain (i.e., 0.02 m). This indicates the mesh was high quality and appropriate for this study.

*4.2. Formulating the Risk Assessment Criteria*

A series of simulations were performed with fire sizes ranging from 200 kW to 1 MW and air cavities ranging from 50 to 200 mm between the insulation and the aluminium panel. The details of all the cases studied are listed in Table 1. The simulation time for all the cases was 200 s.

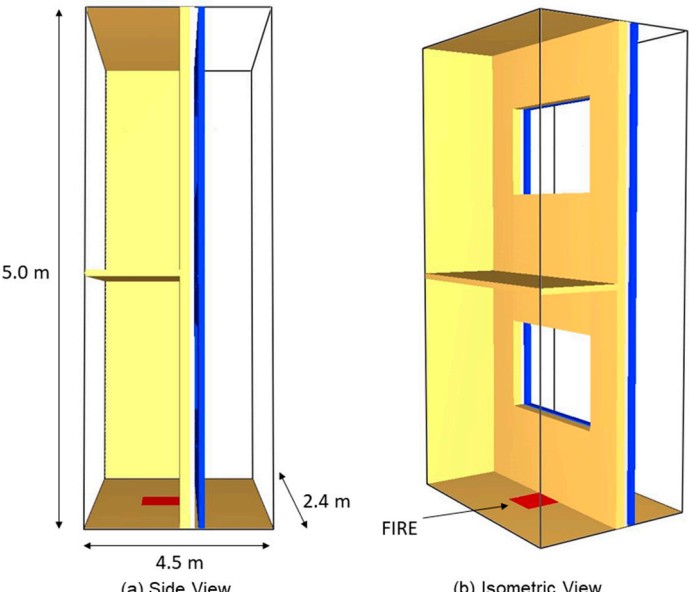

**Figure 5.** Computation domain of the exterior cladding system.

**Table 1.** Numerical simulation cases and configurations for fire size and air cavity.

| Case Number | Initial Fire Size (kW) | Air Cavity |
|:-----------:|:----------------------:|:----------:|
| 1 | 300 | 50 mm |
| 2 | | 100 mm |
| 3 | | 150 mm |
| 4 | | 200 mm |
| 5 | 400 | 50 mm |
| 6 | | 100 mm |
| 7 | | 150 mm |
| 8 | | 200 mm |
| 9 | 500 | 50 mm |
| 10 | | 100 mm |
| 11 | | 150 mm |
| 12 | | 200 mm |
| 13 | 1000 | 50 mm |
| 14 | | 100 mm |
| 15 | | 150 mm |
| 16 | | 200 mm |

The simulations were analysed in terms of ignition of the external cladding system and the tenability limits according to Clause G6.8 in the Code of Practice for Fire Safety in Buildings (AS 4391-1999). The scenario was regarded as untenable at each level when the following conditions were met at the window openings:

1. Radiation > 2.5 kW/ m$^2$
2. CO concentration > 1000 ppm
3. Visibility < 10 m
4. Hot gas temperature > 60 °C.

If either one of the above conditions was reached, the condition was considered untenable. The tenability criteria for each level were determined by firstly, taking the 10 second time average data at the end of each simulation (i.e., from 190 s–200 s) from all the measurement points (i.e., 1A–C for level 1 and 2A–C for level 2); Subsequently, a single value was determined by taking the average of the three measurement point at each level. Figures 6 and 7 shows examples of a best-case scenario where there was no ignition and a worst case where the fire spread over the entire external cladding.

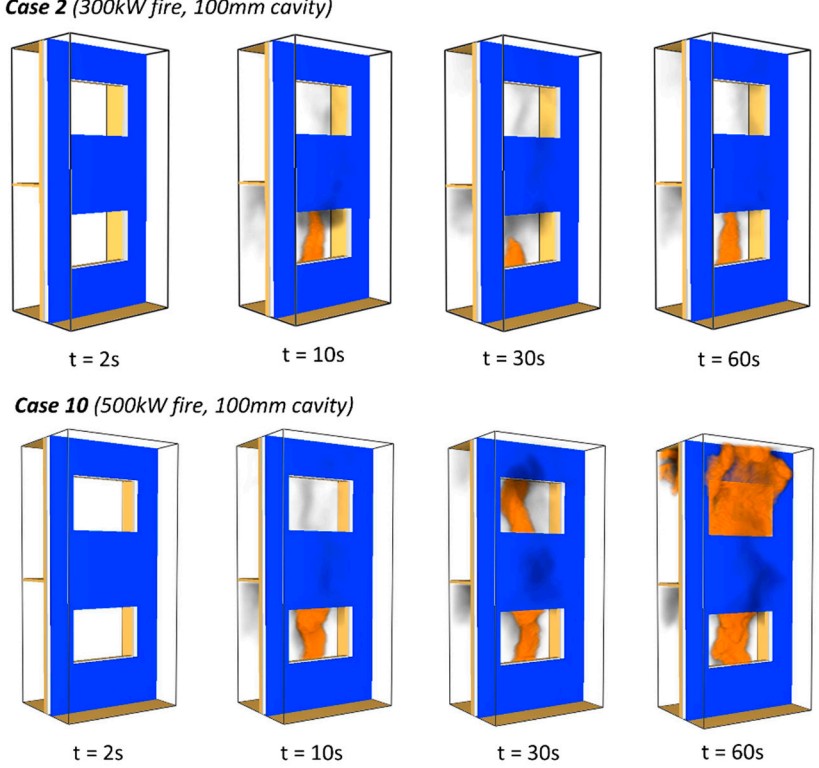

**Figure 6.** Heat release rate (HRR) and smoke rendering of Case 2 where there was no ignition and Case 10 where the fire spread over the entire external cladding.

**Figure 7.** Temperature contour of Case 2 where there was no ignition and Case 10 where the fire spread over the entire external cladding.

## 5. Results

Tables 2 and 3 show the simulation predictions for tenability criteria outlined in Section 4.2 for different fire scenarios. The untenable criteria are highlighted in orange and the tenable criteria in green. As shown in the results, any initial fire sizes higher than 500 kW will result in the ignition of the aluminium composite panel. Decreasing the air cavity resulted in a lower ignition criterion for the composite panel.

**Table 2.** Numerical prediction for the tenability parameters at the first floor for cases with variable initial fire size and air gap between insulation layer and external cladding. (parameters above the untenable criteria are highlighted in orange and those below the criteria are highlighted in green).

| Initial Fire Size (kW) | Air Cavity | Temperature (°C) | Radiation (kW/m$^2$) | CO Concentration (kg/kg) | Visibility (m) |
|---|---|---|---|---|---|
| 300 | 50 mm | 124.088 | 9.599 | 0.008 | 26.955 |
| | 100 mm | 121.000 | 9.260 | 0.009 | 27.400 |
| | 150 mm | 105.384 | 9.019 | 0.006 | 28.500 |
| | 200 mm | 136.248 | 10.695 | 0.002 | 26.397 |
| 400 | 50 mm | 188.248 | 16.256 | 0.013 | 23.825 |
| | 100 mm | 157.000 | 12.300 | 0.011 | 24.200 |
| | 150 mm | 151.965 | 12.453 | 0.011 | 23.900 |
| | 200 mm | 150.959 | 11.734 | 0.010 | 25.302 |
| 500 | 50 mm | 264.833 | 20.267 | 0.025 | 18.300 |
| | 100 mm | 227.962 | 17.919 | 0.019 | 19.561 |
| | 150 mm | 212.563 | 15.087 | 0.017 | 19.400 |
| | 200 mm | 906.114 | 141.943 | 0.109 | 10.200 |
| 1000 | 50 mm | 702.234 | 65.800 | 0.050 | 10.406 |
| | 100 mm | 834.545 | 80.400 | 0.130 | 14.169 |
| | 150 mm | 874.652 | 86.700 | 0.058 | 12.300 |
| | 200 mm | 897.200 | 104.838 | 0.105 | 11.500 |

**Table 3.** Numerical prediction for the tenability parameters at the second floor for cases with variable initial fire size and air gap between insulation layer and external cladding. (parameters above the untenable criteria are highlighted in orange and those below the criteria are highlighted in green).

| Initial Fire Size (kW) | Air Cavity | Temperature (°C) | Radiation (kW/m$^2$) | CO Concentration (kg/kg) | Visibility (m) |
|---|---|---|---|---|---|
| 300 | 50 mm | 42.454 | 0.032 | 0.003 | 23.514 |
| | 100 mm | 42.700 | 0.033 | 0.004 | 27.200 |
| | 150 mm | 43.461 | 0.031 | 0.004 | 28.400 |
| | 200 mm | 48.397 | 0.069 | 0.004 | 24.752 |
| 400 | 50 mm | 203.498 | 18.840 | 0.027 | 8.847 |
| | 100 mm | 51.500 | 0.123 | 0.004 | 25.700 |
| | 150 mm | 60.277 | 0.139 | 0.005 | 28.300 |
| | 200 mm | 61.648 | 0.179 | 0.005 | 24.047 |
| 500 | 50 mm | 673.667 | 60.300 | 0.090 | 6.720 |
| | 100 mm | 809.770 | 82.867 | 0.104 | 8.663 |
| | 150 mm | 829.619 | 56.373 | 0.165 | 3.838 |
| | 200 mm | 847.155 | 89.668 | 0.106 | 4.959 |
| 1000 | 50 mm | 965.972 | 86.100 | 0.140 | 2.690 |
| | 100 mm | 860.300 | 85.823 | 0.108 | 2.701 |
| | 150 mm | 896.000 | 95.800 | 0.139 | 2.793 |
| | 200 mm | 905.370 | 83.249 | 0.095 | 2.610 |

This trend was in agreement with other studies conducted on the effects of cavities on fire dynamics [35,36]. There are three mechanism which enhanced the flammability in a cavity compared to a normal surface: (i) increase radiative heat transfer because of the cavity, (ii) increase upward flame spread from the chimney effect, and (iii) decrease in convective cooling from external air causing an extension of the flame height inside the cavity.

On the other hand, it was observed that a larger air cavity resulted in a higher gas temperature and CO concentration if the cladding was ignited. It may be due to the increased supply of oxygen and area for the fire to develop. This presents a trade-off between initial ignition criteria and the fire size when it was ignited. The results suggest that the heat release rate and flammability of the external

cladding system constructed in this study was significantly influenced by the width of the cavity. Furthermore, there was a threshold or a cavity width which minimised the increase in flammability caused by the cavity while maintaining tenable conditions. However, it is important to note that there are many other factors that influence the overall flammability that were not considered in this study. The results would also be different under different cladding configurations and fire conditions. More case studies need to be examined to achieve better understanding of the flaming behaviour of external cladding systems. Nevertheless, the results presented in this study demonstrate that the proposed simulation framework can be utilised to provide more insight into the fire risk of external cladding systems.

The results highlight one of the significant advantages of computer simulations. All the relevant information such as temperature, CO, and smoke can be extracted from any location in the domain. In addition, it is relatively straightforward to change the configuration of the simulation compared to full-scale experiments. Different factors such as thickness, location, and configurations of the flammable materials can be tested in a simulation environment to provide valuable insight into the ignitability, fire spread, and emissions of non-compliant building materials. A comprehensive database of parameters and consequences for building fires can be formed which can then be processed by predictive models such as artificial neural networks (ANNs) to make forecasts and predictions to the fire risks of a building based on a set of input variables. A systematic assessment tool to evaluate fire risks that is applicable for all non-compliant materials on buildings and provides an indicative grading system to demonstrate the hazardous levels will be highly valuable to the community. This tool can potentially be further developed to incorporate the fire control strategies developed thereby creating a robust response strategy system based on the risks identified. All the above points will be investigated in our future works.

## 6. Conclusions

In the past decade, there has been an increased number of fire incidents associated with combustible cladding materials, especially aluminium composite panels. Many of the severe cases have resulted in catastrophic damages in terms of human casualties, financial losses, and building damage. These catastrophic incidents have raised awareness and concerns by the public regarding non-compliant materials and put immense pressure on government and commercial entities to act on the on the risks associated with the non-compliant building structures. There has been consistent rectification of the relevant building regulations in various jurisdiction worldwide. Responding to this significant new challenge, the Australian Government in 2015 commissioned an ongoing Senate inquiry that has identified non-conforming materials that do not meet the fire safety regulatory standards and pose major fire risks. However, the issue of non-compliant materials in current existing buildings remains. It is recommended that fire engineers develop a greater understanding of the potential fire risks being exposed to provide a more informed assessment of these building facades' combustible performance.

In this article, past fire incidents involving combustible composite panels and the current provisional regulations have been reviewed. A systematic risk assessment tool to analyse non-compliant buildings was formulated. Utilising numerical simulations with a computational fluid dynamic (CFD) approach, different configurations of the flammable materials were tested in a simulation environment to provide valuable insight into the ignitability, fire spread, and emissions of a typical building external cladding facade. Through both literature review of previous large fire incidents, as well as the aid of numerical fire simulations, the tenability criterions for cladding fire scenarios have been extensively studied according to the Australian Fire Safety Code of Practice. The results suggest that any fire sizes higher than 500 kW will result in the ignition of the external cladding. Furthermore, a trade-off has been identified regarding the air cavity between the exterior cladding and the wall. A decrease in air cavity will result in a lower ignition criterion for the external cladding while increasing the cavity results in a larger fire if the material is ignited.

**Author Contributions:** Conceptualization, G.H.Y. and A.C.Y.Y.; methodology, A.C.Y.Y. and G.H.Y.; software, T.B.Y.C. and Q.N.C.; formal analysis, T.B.Y.C.; investigation, T.B.Y.C.; writing—original draft preparation, T.B.Y.C.; writing—review and editing, A.C.Y.Y. and W.Y.; visualization, T.B.Y.C. and Q.N.C.; supervision, G.H.Y. and W.Y.; project administration, A.C.Y.Y.; funding acquisition, G.H.Y.

**Funding:** The article is supported by the Australian Research Council (ARC Industrial Transformation Training Centre IC170100032), the Australian Government Research Training Program Scholarship and the Tactical Research Fund, Bushfire and Natural Hazard Cooperative Research Centre in Australia. All financial and technical supports are deeply appreciated by the authors.

**Conflicts of Interest:** The authors declare no conflict of interest.

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
