# Peer review of "Fire Risk Assessment of Combustible Exterior Cladding Using a Collective Numerical Database"

_fire, doi:10.3390/fire2010011_

Round 1

Reviewer 1 Report

This article presents a systematic risk assessment tool to analyze non-compliant buildings. The study itself makes a lot of sense, but this article is not good enough. My comment is as follows

1.The author said that a database of fire events involving non-compliant building products was developed in this paper, then how to reflect the database established in the article, please elaborate.

2.The article uses a lot of space to review past fire cases, which should be summarized and refined. There is no need to detail each case.

3.More details about the simulation should be provided, such as fire source settings, grid settings, model validation, boundary conditions, etc.

4.Tables 2 and 3 show the simulation predictions for tenability criteria outlined in section 5.3 for 314 different fire scenarios.Where is section 5.3?

5. The data in Table 2 and Table 3 are obtained at which position of the simulation model, it is the mean of the steady state, or the value of a certain moment? Generally, for the case where the dangerous parameter exceeds the critical value, we are more concerned about when the dangerous parameter reaches the critical value.

6. In section 6“The results suggest that any fire sizes higher than 500 kW will result in the ignition of the external cladding.” This sentence is not rigorous enough. This conclusion is only applicable to the simulation of this paper. Whether it can ignite the external cladding is closely related to many factors such as the location of the fire source and the size of the building space.

Author Response

Thank you very much for your valuable suggestions and comments for the paper. The manuscript has been thoroughly revised and the following responses or answers to the comments are provided by the authors accordingly:

Reviewer 1:

This article presents a systematic risk assessment tool to analyze non-compliant buildings. The study itself makes a lot of sense, but this article is not good enough. My comment is as follows

Comment 1:

1.The author said that a database of fire events involving non-compliant building products was developed in this paper, then how to reflect the database established in the article, please elaborate.

Response 1:

The proposed methodology for establishing the fire database was to include both data records and observations from past fire events and simulation results. The data from past fire events was used to construct the simulation cases (e.g. geometry configurations, materials choices). Afterwards, the simulation was used to examine case studies under different fire intensities and cavity sizes between the insulation and exterior composite panel to provide insight on how these factors influence the fire risks. A more comprehensive descriptions of establishing the fire database have been added to the beginning of Section 4.0 Methodology.

Comment 2:

2.The article uses a lot of space to review past fire cases, which should be summarized and refined. There is no need to detail each case.

Response 2:

Section 2 Review of past fire cases have been revised to be more concise and some of the minor cases have been removed.

Comment 3:

3.More details about the simulation should be provided, such as fire source settings, grid settings, model validation, boundary conditions, etc.

Response 3:

A more comprehensive description of the computational model that cover the have details listed in comment 3 been added to Section 4.1. New figure has been added to illustrate the computational domain and the boundary conditions. Details of the mesh and a grid analysis have also been added to the manuscript.

Comment 4:

4. “Tables 2 and 3 show the simulation predictions for tenability criteria outlined in section 5.3 for 314 different fire scenarios.” Where is section 5.3?

Response 4:

The reference was an error and have been revised from “outlined in section 5.3” to “outlined in section 4.2

Comment 5:

5. The data in Table 2 and Table 3 are obtained at which position of the simulation model, it is the mean of the steady state, or the value of a certain moment? Generally, for the case where the dangerous parameter exceeds the critical value, we are more concerned about when the dangerous parameter reaches the critical value.

Response 5:

New description of how the data from Table 2 and Table 3 were determined have been added the revised manuscript.

The locations of the data measurement points have been added to Figure XX. There are three measurement points at the window of each floor. In summary, the tenability criteria for each level were determined by firstly, firstly, taking the 10 second time average data at the end of each simulation (i.e. from 190s-200s) from all the measurement locations (i.e. 1A-C for level 1 and 2A-C for level 2); then take the average of the three measurement location at each level to arrive at a single value for each tenability criteria. 

Comment 6:

6. In section 6“The results suggest that any fire sizes higher than 500 kW will result in the ignition of the external cladding.” This sentence is not rigorous enough. This conclusion is only applicable to the simulation of this paper. Whether it can ignite the external cladding is closely related to many factors such as the location of the fire source and the size of the building space.

Response:

The authors agree with the reviewer that there are many factors that effect the ignitability of external claddings. The problem is very complex, and the cavities and fire sizes tested in this study only reflect a small sample of cases. The results section has been revised to be more precise and modest in reporting the key findings from this study and also take into consideration other factors that may influence the fire risks of these cladding panels such as physical barriers in the cavity, insulation material and fire locations.

Reviewer 2 Report

The paper is well written, and overall has great merit. The findings in chapter 3 (so many buildings not meeting the BSA) are surprising. It is an appreciated effort to investigate the topic of the facade fires, and I would recommend publishing this paper after a major revision.

It is due to the great impact the topic of the facades has on fire community, we all need to temper down some of our beliefs and "shocking" statements. I would ask the Authors to not use statements such as "all fires (...) have been investigated" or that combustibility of material is the sole root of the problem. This paper may be a worty element of the discussion, but it does not close the discussion, nor solve the facade problem. Please choose the statements accordingly (I write more on that below).

Major comments:

- l. 36, as I had a chance (it was not a pleasure...) to test a "Grenfell" type facade in a laboratory (outside UK) in full scale, I would not be so certain that the polymer materials are the sole cause of the problem. It may be a combination of the cavity, bad barries in the cavity, higly insulative material etc. There was a recent paper by M. Bonner and G. Rein (http://ctbuh-korea.org/ijhrb/05ijhrb01.php?id=244#) related to the complexity of the problem. The topic is very interesting, but you cannot claim you solved it by pointing to combustability (extraordinary claim requires extraordinary proof). Please rework the introduction to point to possible causes, that not only lie in combustability. Especially that further in the text you refer to cavities. 

- It is not defined in the text, but I think your definition of cladding panels is narrow to rainscreen type composite panels. However, in major parts of the world (northern and central europe as an example) the main combustible claddings are simple layered materials for thermal insulation (various types of ETICS with EPS+mortar, or sandwitch panels with PUR/PIR cores). These types of cladding do not have the same problems as the one you describe. I would like to ask you to somehow identify this in the introduction, and the review of past fires.

-  l.258 any validation case for combustion in a narrow gap in high velocity? I don't think I recall such case in the FDS validation book, but may be wrong.

- l. 272 why such a low value of Y_soot was chosen? The soot concentration value is taken by the FDS to solver "RADCAL" which determines the emmissivity of smoke within every discrete cell. This affects radiation (heavily!). In the case of the gap within facade, radiation may have a significant effect. Was any sensitivity study performed for this?

- l. 276 I am pretty sure that FDS developers consider constant Smag model as not validated anymore, and was replaced with more efficient solutions (Deardorf). Have you ran any sensitivity test with different variants of LES?

- there is no information regarding mesh used, please include that.

- l. 329 - to form a neural network you need much more data. And validated data on top of that! You cannot be sure that FDS solved everything (especially with the limitations in the pyrolysis models). 

Minor comments:

- the statement in l. 19: "(...) all of the relevant major fire events in Australia and other countries(...)" is too strong. I would appreciate softening it down to "multiple" or "most of"

- l.60 - was this really the first external cladding fire in the history of world? Please use modest terms, like "first fire investigated for this study was (...)".

- are photographs in the text under CC BY license? Please indicate the rights and authorship accordingly for each, taken into account the Fire journal is an open access under CC BY.

- l. 248 As I have a chance to run a large fire laboratory and do CFD, I don't think I completely agree. The full scale research is paramount to understand what is happening in the facade. However, the (validated) CFD is valuable counterpart in parametric research, to explore the system sensitivity to some of the parameters (like gap width)

Author Response

Thank you very much for your valuable suggestions and comments for the paper. The manuscript has been thoroughly revised and the following responses or answers to the comments are provided by the authors accordingly:

Reviewer 2:

The paper is well written, and overall has great merit. The findings in chapter 3 (so many buildings not meeting the BSA) are surprising. It is an appreciated effort to investigate the topic of the facade fires, and I would recommend publishing this paper after a major revision.

It is due to the great impact the topic of the facades has on fire community, we all need to temper down some of our beliefs and "shocking" statements. I would ask the Authors to not use statements such as "all fires (...) have been investigated" or that combustibility of material is the sole root of the problem. This paper may be a worty element of the discussion, but it does not close the discussion, nor solve the facade problem. Please choose the statements accordingly (I write more on that below).

Response:

The authors agree with the comment. The manuscript was too ambitious with statements that were too strong. The manuscript has been thoroughly revised to be more modest and precise and take into consideration all the recommendations provided by the reviewer.

Major comments:

Comment 1:

- l. 36, as I had a chance (it was not a pleasure...) to test a "Grenfell" type facade in a laboratory (outside UK) in full scale, I would not be so certain that the polymer materials are the sole cause of the problem. It may be a combination of the cavity, bad barries in the cavity, higly insulative material etc. There was a recent paper by M. Bonner and G. Rein (http://ctbuh-korea.org/ijhrb/05ijhrb01.php?id=244#) related to the complexity of the problem. The topic is very interesting, but you cannot claim you solved it by pointing to combustability (extraordinary claim requires extraordinary proof). Please rework the introduction to point to possible causes, that not only lie in combustability. Especially that further in the text you refer to cavities.

Response 1:

The introduction has been revised to take into consideration other factors which influence the fire risks of external cladding systems.

Comment 2:

- It is not defined in the text, but I think your definition of cladding panels is narrow to rainscreen type composite panels. However, in major parts of the world (northern and central europe as an example) the main combustible claddings are simple layered materials for thermal insulation (various types of ETICS with EPS+mortar, or sandwitch panels with PUR/PIR cores). These types of cladding do not have the same problems as the one you describe. I would like to ask you to somehow identify this in the introduction, and the review of past fires.

Response 2:

In Australia, we often use the phrases “external wall cladding” and “aluminium sandwich panel” interchangeably. The introduction has been revised include more detail description of External Thermal Insulation Composite Systems (ETICS) which typical consist of a thermal insulation layer and a surface finish layer. And the key risk factor lies in the use highly combustible aluminium sandwich panels which causes rapid fire spread at the exterior wall. In addition, The manuscript have been revised to change some instances of the word “cladding” to “sandwich panels or composite panels” to avoid misunderstanding.

Comment 3:

-  l.258 any validation case for combustion in a narrow gap in high velocity? I don't think I recall such case in the FDS validation book, but may be wrong.

Response 3:

To the best of our knowledge there are no validation cases for combustion in narrow gaps in high velocity. But there are façade test data that can be used to provide validation. This study provides a preliminary study to investigate the viability of our proposed framework for the fire risk assessment tool.  In our future work, we plan to conduct fire test to provide validation for our simulations.

Comment 4:

- l. 272 why such a low value of Y_soot was chosen? The soot concentration value is taken by the FDS to solver "RADCAL" which determines the emmissivity of smoke within every discrete cell. This affects radiation (heavily!). In the case of the gap within facade, radiation may have a significant effect. Was any sensitivity study performed for this?

Response 4:

The authors agree that soot contribute significantly in radiative heat transfer especially in absorption coefficients. The value of 0.01 for the soot yield based on the reaction of polyurethane (GM37) from the SFPE Handbook. The authors also agree that the 0.01 is a conservative value, however only one reaction fuel can be active in FDS simulations which is also applied to the square burner. The soot yield parameter was previously studied comprehensively by Vigne and Węgrzyński:

1.         Influence of Variability of Soot Yield Parameter in assessing the safe conditions in Advanced Modelling Analysis. Results of Physical and Numerical Modelling Comparison, 11th SFPE Conference on Performance-Based Codes and Fire Safety Design, Warsaw, 2016.

A wide range of soot yields ranging from 0.001 g/g to 0.178 g/g were tested to replicated various types of flames. It was discovered that a range from 0.01 g/g to 0.015 g/g was suitable for polymeric composite materials such as polyethylene (PE), which is a similar material applied in the current study. This part of the discussion for the choice of the value 0.01 g/g for soot yield has been now included in the revised manuscript.

Comment 5:

- l. 276 I am pretty sure that FDS developers consider constant Smag model as not validated anymore, and was replaced with more efficient solutions (Deardorf). Have you ran any sensitivity test with different variants of LES?

Response 5:

To the best of our knowledge, the Smagorinsky model is the only LES model available in FDS and the other option is DNS. Therefore, we did not perform any sensitivity test for different LES models. In addition, the Smagorinsky model is still widely adopted for compartment fire simulations and there are many validation studies done on this topic.

Comment 6:

- there is no information regarding mesh used, please include that.

Response 6:

A more comprehensive description of the computational domain, mesh and boundary conditions have been added to Section 4.1.

Comment 7:

- l. 329 – to form a neural network you need much more data. And validated data on top of that! You cannot be sure that FDS solved everything (especially with the limitations in the pyrolysis models).

Response 7:

The authors agree that more data is needed to form a neural network. This study provides a preliminary study to investigate the viability of our proposed framework for the fire risk assessment tool.  In our future work, more simulations case studies will be performed to examine other factors such as different materials, barriers in the cavity, fire locations. Furthermore, we are also planning to conduct both benchtop scale fire test (TGA, Cone Calorimeter) and full-scale façade test to contribute to the database and provide validation data.  The goal of this project is to gather enough data to form an ANN which can be used to provide insight into the fire risks of aluminium composite panels.

Minor comments:

- the statement in l. 19: "(...) all of the relevant major fire events in Australia and other countries(...)" is too strong. I would appreciate softening it down to "multiple" or "most of"

Response:

The statement has been revised from “all of the relevant major fire events” to “review of relevant major fire events...

- l.60 - was this really the first external cladding fire in the history of world? Please use modest terms, like "first fire investigated for this study was (...)".

Response:

The statement have been revised from “The first historical case..” to “One of the first documented cases of…”

- are photographs in the text under CC BY license? Please indicate the rights and authorship accordingly for each, taken into account the Fire journal is an open access under CC BY.

Response:

All the photographs that are not under the CC BY license have been removed.

- l. 248 As I have a chance to run a large fire laboratory and do CFD, I don't think I completely agree. The full scale research is paramount to understand what is happening in the facade. However, the (validated) CFD is valuable counterpart in parametric research, to explore the system sensitivity to some of the parameters (like gap width)

Response:

The authors agree that both full scale fire tests and CFD studies are valuable counterparts in research. The statement on line 248 at the beginning of Section 4 Methodology have been revised.

Round 2

Reviewer 2 Report

Dear Author, 

I am satisfied with the changes done to the manuscript, and comments provided. I have some final minor remarks:

- For the paper on soot yield, there was a journal article which should be a better citation than conference materials (https://www.sciencedirect.com/science/article/pii/S0379711217301327). I also think you should say that the authors "reported" these values and not "discovered" them, as I understand they summarized the previous research.

- Citation [29] must be corrected, you gave link to the editorial of SFPE Handbook, while you had in mind some particular chapter of the book. I think you meant this chapter (and this is the most recent issue):

McGrattan, K., Miles, S., 2016. Modeling Fires Using Computational Fluid Dynamics (CFD), in: SFPE Handbook of Fire Protection Engineering. Springer New York, New York, NY, NY, pp. 1034–1065. https://doi.org/10.1007/978-1-4939-2565-0_32

- regarding constant Smagorinsky i meant sub model used to determine C_smag value (which there are 3 sub models to choose from in FDS). However, following your uptade to the mesh chapter, I do not find this discussion relevant that much and this does not require further actions.

Author Response

The authors appreciate all the valuable suggestions and comments for the manuscript given by reviewer. The manuscript has been carefully read through to remove all grammatical and typographical errors. The following are the author’s responses to the reviewer’s comments:

Reviewer comments:

I am satisfied with the changes done to the manuscript, and comments provided. I have some final minor remarks:

Comment 1:

- For the paper on soot yield, there was a journal article which should be a better citation than conference materials (https://www.sciencedirect.com/science/article/pii/S0379711217301327). I also think you should say that the authors "reported" these values and not "discovered" them, as I understand they summarized the previous research.

Response 1:

Additional references have been added to the discussion on soot yields. In addition, the word “discovered” have been changed to “reported” in line 276.

Comment 2:

- Citation [29] must be corrected, you gave link to the editorial of SFPE Handbook, while you had in mind some particular chapter of the book. I think you meant this chapter (and this is the most recent issue):

McGrattan, K., Miles, S., 2016. Modeling Fires Using Computational Fluid Dynamics (CFD), in: SFPE Handbook of Fire Protection Engineering. Springer New York, New York, NY, NY, pp. 1034–1065. https://doi.org/10.1007/978-1-4939-2565-0_32

Response 2:

Citation [29] have been changed to the one recommended by the reviewer. The original citation was a reference to the characteristic length scale D* expression which incorporates the heat release rate Q of the fire. The recommended reference is much more relevant and direct. Furthermore, an additional sentence have been added to on line 304 to outline how R* is calculated.

Comment 3:

- regarding constant Smagorinsky i meant sub model used to determine C_smag value (which there are 3 sub models to choose from in FDS). However, following your uptade to the mesh chapter, I do not find this discussion relevant that much and this does not require further actions.

Response 3:

Thank you very much for clarification and sorry for misunderstanding the comment. We were not fully aware of the list of submodels that have been implemented into recent versions of FDS. We will definitely take these factors into consideration in future studies.
